# Enforcing Axioms for AI Alignment under Loss-Based Rules

**Alexandros Hollender**[*]
University of Oxford

alexandros.hollender@cs.ox.ac.uk

**Sonja Kraiczy**[*]
University of Oxford,
PIBBSS[†]
Sonja.Kraiczy@cs.ox.ac.uk

## Abstract

Recent alignment methods for large language models, most notably reinforcement learning from human feedback (RLHF), often train an auxiliary reward model to minimize a loss function on binary preference data over model responses. We study a theoretical setting inspired by principle-guided methods such as Constitutional AI, in which a small set of principles (e.g., helpfulness, toxicity) act as "voters" that guide binary comparisons—such as preferring the less toxic response. We model these principles as linear directions in an embedding space of responses, a simplifying assumption motivated by the Linear Representation Hypothesis—concepts are linear directions in representation-space—a useful first-order approximation in practice. In this *linear social choice model*, Ge et al. (2024) showed that an optimal linear reward model can violate Pareto optimality (PO): From the principles-as-voters lens, this means a response A can be less helpful and more toxic than B, yet still receive a higher reward. We analyze axiomatic violations in the linear social choice setting and probe the robustness of negative results under realistic assumptions. We show that added expressivity does not resolve the issue: polynomial reward models can still fail PO. We then offer a pragmatic alternative showing that when the data uniformly covers the embedding space, broad classes of loss-based rules in the limit exactly recover the axiomatic guarantees. This yields a recipe for constitutional-style alignment with provable guarantees: enforce balanced coverage *via dataset design* to restore axiomatic guarantees without abandoning standard training pipelines.

## 1 Introduction

Many recent alignment methods follow a common pipeline: collect pairwise (binary) preferences over model responses, fit a reward (or preference) model to these comparisons, and then optimize the base model to minimize a loss correlated with aligned behavior. In the classical human-in-the-loop setting of Reinforcement Learning from Human Feedback (RLHF) (Christiano et al., 2017), human annotators select the preferred response between candidate responses. Rather than eliciting unguided preferences, Anthropic introduced principle-oriented feedback (Bai et al., 2022a) in which annotators judge responses against explicit principles—helpfulness, honesty, and harmlessness (HHH); this principle-guided supervision later informed Constitutional AI, which formalizes a written set of principles to guide binary comparisons of model responses (Bai et al., 2022b). We use "constitutional-style" as an umbrella term for such principle-guided supervision.

Adopting a social-choice perspective on constitutional-style alignment, we treat principles as voters that evaluate pairs of model responses by judging which better adheres to the relevant principle. From this perspective, a minimal requirement - "axiom" - for any aggregation method is Pareto Optimality (PO) (also known as Unanimity (Arrow, 1951)):

---

[*]Authors are listed in alphabetical order.
[†]PIBBSS (Principles of Intelligence).

e.g., in the HHH framework, if response A is more helpful, more honest, and more harmless than B, then A should receive higher reward and thus be more likely to be generated by the model. We study a setting in which principles (such as HHH) are modeled as linear directions in a representation space. This is a simplifying assumption motivated by the Linear Representation Hypothesis (LRH) (Park et al.), which posits that some high-level concepts are well-approximated by linear features in learned embeddings and which has been operationally useful in mechanistic interpretability.

Formally, most of our results lie within the *linear social-choice* framework of Ge et al. (2024): utilities over a fixed representation are linear, pairwise comparisons yield binary labels, and optimizing those labels learns a linear reward model. Although the linear social-choice framework was introduced as a model for RLHF, we contend that both the linearity assumption and axioms such as PO are more realistic and salient in constitutional-style alignment with a small number of principles.[1]

Within the linear social choice model, the authors show that, perhaps surprisingly, an optimally trained linear reward can violate Pareto optimality (PO): everyone prefers response A to response B, yet B receives higher reward. The concrete counterexample involves only two (weighted) principles, thus covering the HHH scenario; we discuss it as a warm-up in Section 3. Finding this result counterintuitive, we ask:

*How robust are axiomatic violations under loss-based training?*

## 1.1 Our Contribution

We revisit the axiomatic violations through a lens which is compatible with training pipelines. As part of our warm-up in Section 3, we provide new intuition on why the PO violation occurs in the first place, by providing a simplified minimal example that clarifies what goes wrong. While Ge et al. (2024) show that a social choice combinatorial approach can recover axiomatic guarantees, our paper focuses on approaches to obtain guarantees that align with modern ML pipelines—namely, loss-based rules. Our main contributions are three-fold.

1. **Beyond linear rewards.** A natural hypothesis is that linear rewards with a frozen embedding are simply too restrictive; perhaps keeping voters linear but using a more expressive reward model, such as bounded-degree polynomials, restores PO. We show that, again surprisingly, this is not the case: even with linear voters and expressive reward models, violations of PO (and of another natural axiom called Pairwise Majority Consistency) persist; see Section 4.

2. **Generalization.** In Section 5, we adopt an axiomatic framework on the embedding space—rather than a fixed train set—and define a generalized version of Pareto Optimality (PO). This recognizes the key feature of the linear model - directional differences between candidates matter, not their embeddings. With finite data, we cannot expect to satisfy these axioms exactly; This aligns with the core role of reward models: their value is generalization, so the key question is how well any desirable property transfers to unseen data.

3. **Data-centric perspective**. While social-choice axiomatic analyses typically reason from a *worst-case perspective*, real-world training dynamics are far more sensitive to how we curate and sample (preference) data. Recognizing the limitation of the worst case approach, we ask: can we *choose* binary comparison queries so that axiomatic guarantees are provably recovered? In Section 5, we show that a random sampling scheme already suffices and in the limit we achieve perfect PO. This suggests practical levers (data inspection, reweighting or careful dataset design) when combinatorial social-choice algorithms are hard to deploy in real systems.

Results marked with (♦) have their proofs deferred to the appendix.

---

[1]With many raters, agreement on a binary comparison becomes increasingly unlikely.

## 1.2 Related Work

The field of social choice (Arrow et al., 2010; Brandt et al., 2016) offers a long tradition of axiomatic guarantees that provide a lens through which to compare different aggregation rules. This perspective is directly relevant to preference-based alignment: Reinforcement Learning from Human Feedback (RLHF) (Christiano et al., 2017) and Nash Learning from Human Feedback (NLHF) (Munos et al., 2024) have natural analogues in social choice. The optimal solution to the Bradley–Terry loss in RLHF is known to induce a ranking that is equivalent to the Borda ranking from social choice theory (Anderson et al., 2009; Siththaranjan et al., 2024). Formally, NLHF coincides with the von Neumann winner defined in Dudík et al. (2015), which in turn matches an old rule from social choice, Fishburn's maximal lotteries (Fishburn, 1984).[2] Building on this connection, several position papers have recently argued for social choice as a useful lens on RLHF and alignment more broadly (Conitzer et al., 2024; Dai & Fleisig).

Among the first wave of technical results that has emerged since, most relevant to us is the work by Ge et al. (2024) who propose the linear social-choice model where voters are linear directions over a fixed embedding. Building on prior work by (Noothigattu et al., 2020) which explores the axiomatic properties of reward functions defined as MLE estimators of underlying random utility models, they analyze the ranking over a fixed candidate set induced by an optimal linear reward with respect to axioms including Pareto optimality (PO) and Pairwise Majority Consistency (PMC). Because these axioms can still be violated in that setting, they adapt a combinatorial social-choice rule to the linear social-choice model to obtain the guarantees. The closest analogue in social choice to linear preferences over a fixed embedding arises in the literature on restricted preference domains (over rankings) (Elkind et al., 2022).

Procaccia et al. (2025) analyze clone robustness and show that an appropriate reweighting of the Bradley–Terry loss can be made to satisfy their axiom. Most existing work at the intersection of alignment and social choice adopts the classic social-choice setting with unstructured alternatives, in contrast to the metric setting we study here. Recent work in this vein includes representative social choice proposed by Qiu (2024), the extension of the distortion framework of Procaccia & Rosenschein (2006) to preference distributions by Gölz et al. (2025), an analysis of RLHF when each comparison is labeled by a single annotator by Xiao et al. (2025), and a proposal towards proportional alignment by Kim et al. (2025).

## 2 Preliminaries

**Social choice model.** The set of alternatives is the $d$-dimensional space $\mathbb{R}^d$. A reward function $r : \mathbb{R}^d \to \mathbb{R}$ induces a (weak) ordering $\preccurlyeq_r$ over the set of all alternatives $\mathbb{R}^d$ by ranking alternatives according to their rewards, namely $a \preccurlyeq_r b \iff r(a) \leq r(b)$ for all $a, b \in \mathbb{R}^d$.

There are $n$ voters, and each voter $i$ has a (weak) ordering $\preccurlyeq_i$ over the set of alternatives $\mathbb{R}^d$. Unlike in the usual social choice setting, we do not observe the full orderings, but instead we are only offered a partial view of each voter's preferences. For each voter $i$, we observe a list of pairwise (strict) comparisons. Formally, for each $i$ we are given a nonempty finite set $P_i \subset (\mathbb{R}^d)^2$ that satisfies $(a, b) \in P_i \implies a \prec_i b$. A natural special case is when there is a set $C$ of $m$ candidates in $\mathbb{R}^d$ and the sets $P_i$ contain all[3] pairwise comparisons between candidates in $C$, i.e., $P_i = \{(a, b) \in C^2 : a \prec_i b\}$.

In this context, a voting rule takes as input sets of pairwise comparisons $P_1, \ldots, P_n$, one for each voter, and outputs a reward function whose induced ordering attempts to aggregate the voter preferences.

---

[2]Both concepts are Nash equilibria of games and coincide because the corresponding payoff matrices are related by a positive affine transformation; see (Wang et al., 2023; Maura-Rivero et al., 2025) for the explicit reference to maximal lotteries.

[3]We assume that the set of candidates $C$ is such that no two candidates are tied for any of the $n$ voters, i.e., for any $a, b \in C$ either $a \prec_i b$ or $b \prec_i a$ holds.

**Loss-based voting rules.** We restrict our attention to voting rules that output a reward function minimizing a total loss function. Given a particular loss function $\ell : \mathbb{R} \to \mathbb{R}$, we define the total loss incurred by reward function $r$ as

$$\mathcal{L}(r) := \sum_{i \in [n]} \sum_{(a,b) \in P_i} \ell(r(a) - r(b))$$

where $P_1, \ldots, P_n$ are the sets of pairwise comparisons of the $n$ voters, as described above. The voting rule defined by loss function $\ell$ outputs a reward function $r$ minimizing the total loss $\mathcal{L}$.

We obtain the Bradley-Terry loss (Bradley & Terry, 1952; Zermelo, 1929) $\mathcal{L}_{BT}$ used in RLHF by choosing the loss function $\ell$ to be the cross-entropy loss, i.e., $\ell_{\mathrm{BT}}(x) := \log\big(1 + e^x\big)$.

We will mention the precise assumptions on the loss function $\ell$ in the theorem statements later, but for now let us just think of $\ell$ as a strictly increasing function. Then the idea is that if some voter has ranked $a$ over $b$ in their set $P_i$, a reward function should be penalized for ranking $b$ over $a$ (i.e., for assigning a higher reward to $b$ than $a$).

**Linear social choice.** We will mostly focus on the linear social choice model, where the reward functions are linear, i.e, of the form $r_\theta(x) = \langle \theta, x \rangle$. In particular, in this context, a loss-based voting rule will minimize the total loss $\mathcal{L}$ over the set of all linear reward functions. Furthermore, the voters are also assumed to have orderings that are induced by linear reward functions.

**Axioms.** In this paper we will focus on the following two natural axioms.

**Definition 1** (Pareto Optimality (PO)). A loss-based voting rule satisfies Pareto optimality (PO), if it outputs a reward function $r$ that ranks any comparison $(a, b) \in \cap_i P_i$ correctly, i.e., $r(a) < r(b)$.

In other words, if there exists a comparison between two candidates on which all voters agree, then PO requires the voting rule to also rank these candidates in the same way as the voters.

**Definition 2** (Pairwise Majority Consistency (PMC)). An ordering $\prec$ over all alternatives appearing in the instance is a PMC ordering, if it satisfies: $a \prec b$ if and only if a strict majority of voters rank $a$ below $b$, i.e., $|\{i : (a, b) \in P_i\}| > n/2$. A loss-based voting rule satisfies PMC, if whenever a PMC ordering $\prec$ exists, it outputs a reward function that induces $\prec$.

Note that if a PMC ordering exists, then it is necessarily unique.

## 3 Revisiting the Pareto Optimality Violation

The counterexample to PO provided by Ge et al. (2024) is constructed in $\mathbb{R}^2$. It uses two (weighted) voters with direction vectors $v_1 = (1, 1)$ and $v_2 = (-1, 0)$ (their magnitudes are irrelevant) and six candidates in $\mathbb{R}^2$, arranged as two triples. The key idea is to place one triple ($a$ at $(2, 1)$, $b$ at $(1, 1)$ and $c$ at $(0, 0)$) so that the two voters disagree on the induced ordering of its three candidates; with suitable weights, however, the optimal direction is $(1, 0)$.

Each candidate is then duplicated and the copies are perturbed within an $\varepsilon$-neighborhood of the originals. This replicates every pairwise comparison fourfold and adds an almost constant term to the loss from comparing each candidate with its own copy. By a generalized continuity argument, for sufficiently small $\varepsilon$ the optimal linear reward for the perturbed instance remains close to the original one, since as $\varepsilon \to 0$ the new loss is (in the limit) a linear transformation of the old loss. Finally, one forces one of these copies, say $c'$, to approach $c = (0, 0)$ from a direction in the left half-space that makes both $v_1$ and $v_2$ strictly prefer $c'$ to $c$, yielding the desired PO violation.

This construction can be tightened to use only four candidates by duplicating a single point, as we do in the proof of Theorem 4.1. With three non-collinear candidates and any number

of voters, PO is always satisfiable: optimal rewards (note that these are only unique up to an additive constant) of the unconstrained objective over these three points can always be realized by some $\theta \in \mathbb{R}^2$. Disagreement between voters (e.g., $v_1$ prefers $a \succ b \succ c$ while $v_2$ prefers the reverse) acts like a *length constraint* in the Bradley–Terry loss. For a pair $(a, b)$, the loss contains both

$$\log\!\big(1 + e^{-\langle \theta, a-b \rangle}\big) \quad \text{and} \quad \log\!\big(1 + e^{-\langle \theta, b-a \rangle}\big),$$

(possibly with different weights). Letting $\|\theta\| \to \infty$ drives exactly one of these two terms to $+\infty$ while the other tends to 0; hence the loss penalizes unbounded $\|\theta\|$. (The same phenomenon holds more generally for losses bounded below but diverging to $\to \infty$ as their argument $\to \infty$.)

We next study a minimal example with one voter and three candidates where the optimal linear reward subject to a norm constraint violates PO. This instance provides insight into why PO can fail even without such explicit constraints on instances with multiple voters. Consider a single voter $v = (\varepsilon, 1)$ with $\varepsilon > 0$ and three candidates $a = (1, 0)$, $b = (0, 0)$, $c = (-\delta, \delta)$ where $0 < \delta \ll 1$. Constrain $\theta$ to unit length, $\theta = (\theta_1, \ \pm\sqrt{1 - \theta_1^2})$ with $\theta_1 \in [-1, 1]$. The pairwise Bradley–Terry terms are

$$(a, b): \ \log\!\big(1 + e^{-\langle \theta, a-b \rangle}\big) = \log\!\big(1 + e^{-\theta_1}\big),$$

$$(a, c): \ \log\!\big(1 + e^{-\langle \theta, a-c \rangle}\big) = \log\!\Big(1 + e^{-\big((1+\delta)\theta_1 \ \pm \ \delta\sqrt{1-\theta_1^2}\big)}\Big),$$

$$(b, c): \ \log\!\big(1 + e^{-\langle \theta, c-b \rangle}\big) = \log\!\Big(1 + e^{\delta(\theta_1 \ \pm \ \sqrt{1-\theta_1^2})}\Big) \ \leq \ \log\!\big(1 + e^{\delta\sqrt{2}}\big),$$

since $\max_{\theta_1 \in [-1,1]}\big(\theta_1 \pm \sqrt{1 - \theta_1^2}\big) = \sqrt{2}$. Thus the $(b, c)$ term is $O(\delta)$, while the $(a, b)$ term strictly decreases as $\theta_1$ increases, and for small $\delta$ the $(a, c)$ term also decreases with $\theta_1$. Consequently, for sufficiently small $\delta$ any minimizer has $\theta_1$ close to 1 and $\theta$ in the upper-right quadrant near the $x$-axis, yielding $\langle \theta, b \rangle > \langle \theta, c \rangle$ while $\langle v, b \rangle > \langle v, c \rangle$. Thus, the presence of other voters can impose a length constraint, and such length constraints lead to PO violations - even for a single voter. Intuitively, the norm constraint can be interpreted as a finite "budget"; directions (such as $a - b$) that are more common or are longer dominate the loss because "misclassification" in such directions contributes larger terms to the loss, and so these terms are prioritized given a length constraint.

## 4 Polynomial Reward Functions

In this section, we study an extension of the linear social choice model, where we allow more general reward functions. Namely, we consider polynomial rewards and show that PO and PMC fail even in this case.

**Theorem 4.1.** *Any loss-based voting rule with a loss function $\ell$ that is strictly convex, lower bounded, and differentiable with $\ell'(0) > 0$, fails to satisfy PO and PMC even with polynomial reward functions of bounded degree. Furthermore, this already holds in two dimensions and with three voters that all lie in the positive quadrant.*

In particular, this theorem applies to the Bradley–Terry loss, which thus fails to satisfy PO and PMC even with polynomial reward functions of bounded degree.

### 4.1 Proof of Theorem 4.1

The instance consists of $m + 1 := d(d + 1) + 2$ candidates $c_0, c_1, c_2, \ldots, c_m$, whose positions in $\mathbb{R}^2$ will be specified later. Furthermore, there are two weighted voters:

- Voter $v_1 = (1, 0)$ has a fraction $\alpha \in (1/2, 1)$ of the votes and ranks the candidates in the order
$$c_1 \prec c_0 \prec c_2 \prec c_3 \prec \cdots \prec c_m.$$
- Voter $v_2 = (0, 1)$ has a fraction $1 - \alpha$ of the votes and ranks the candidates in the order
$$c_m \prec \cdots \prec c_1 \prec c_0.$$

Note that the voters disagree on all comparisons, except the comparison between $c_0$ and $c_1$, where they agree. The proof works for any $\alpha \in (1/2, 1)$, but we can set, e.g., $\alpha = 2/3$ to obtain a setting with three (unweighted) voters.

**Total loss function.** For now, consider allowing arbitrary reward values $r_0, r_1, \ldots, r_m \in \mathbb{R}$ for the candidates $c_0, c_1, \ldots, c_m$. The total loss of the reward vector $r$ on this instance can be written as

$$
\begin{aligned}
\mathcal{L}(r) = \mathcal{L}(r_0, r_1, \ldots, r_m) := \alpha & \left( \ell(r_1 - r_0) + \sum_{j=2}^{m} \ell(r_0 - r_j) + \sum_{1 \leq i < j \leq m} \ell(r_i - r_j) \right) \\
& + (1 - \alpha) \sum_{0 \leq i < j \leq m} \ell(r_j - r_i)
\end{aligned}
\tag{1}
$$

Note that we can assume without loss of generality that, say, $r_1 = 0$, since the total loss does not change if we subtract $r_1$ from all rewards.

**Optimal rewards in the degenerate instance.** In our instance, we will position candidate $c_0$ very close to $c_1$. Thus, the instance will be closely related to a "degenerate" instance, where $c_0$ and $c_1$ lie at the same position. In that degenerate instance, the optimal (arbitrary) rewards are given by the following optimization problem:

$$
\min_{r} \quad \mathcal{L}(r) \quad \text{s.t.} \quad r_0 = r_1 = 0
\tag{2}
$$

**Claim 1 (♦).** *The optimization problem (2) has a unique solution $0 = r_1^* < r_2^* < \cdots < r_m^*$.*

**Positioning of the candidates.** We use these optimal rewards $0 = r_1^* < r_2^* < \cdots < r_m^*$ to define the positions of the candidates. First, we let $c_1 = (0, 0)$ and $c_0 = (\delta, \delta)$ for some sufficiently small $\delta > 0$ to be specified later. Next, for each $j \in [d]$, we let $L_j := \{(x, y) \in \mathbb{R}^2 : y = -2x + j\}$, i.e., $L_j$ is the line going through point $(0, j)$ with slope $-2$. Then, for each $i = 2, 3, \ldots, n$, we position candidate $c_i$ at the unique point $(x, y)$ on line $L_j$ such that $-x - y = r_i^*$, where $j = \lceil (i-1)/(d+1) \rceil$. As a result, for any $i \geq 2$, candidate $c_i$ is positioned at $(x, y) = (r_i^* + j, -2r_i^* - j)$, where $j = \lceil (i-1)/(d+1) \rceil$.

We give a more detailed description of the positioning of candidates $c_i$ for $i \geq 2$. We let $c_2$ be the unique point $(x, y)$ on line $L_1$ such that $-x - y = r_2^*$. Similarly, we let $c_3$ be the unique point $(x, y)$ on line $L_1$ such that $-x - y = r_3^*$. We continue positioning points on $L_1$ in this manner, until $d + 1$ points have been positioned. Then, we switch to $L_2$. In other words, $c_{d+3}$ is the unique point $(x, y)$ on line $L_2$ such that $-x - y = r_{d+3}^*$. We proceed in this manner, placing $d + 1$ points on each line, and then moving on to the next line. Note that the number of candidates to be positioned (without $c_0$ and $c_1$, since these have been fixed above) is exactly $m - 1 = d(d+1)$, so we are able to position exactly $d + 1$ candidates on each of the $d$ lines $L_1, \ldots, L_d$. See Figure 1 for an illustration. By construction the following holds.

**Claim 2.** *The positions of the candidates $c_0, \ldots, c_m$ are consistent with the rankings of voters $v_1$ and $v_2$. Furthermore, the polynomial $p^*(x, y) = -x - y$ satisfies $p^*(c_i) = r_i^*$ for all $i \in [m]$ and ranks the candidates in the order $c_0 \prec c_1 \prec c_2 \prec \cdots \prec c_m$.*

**Degree-$d$ polynomial reward functions.** We consider the class of reward functions that are polynomials of degree at most $d$. As before, without loss of generality, we can restrict our attention to polynomials $p$ that satisfy $p(c_1) = 0$, i.e., the constant term is zero (since $c_1$ lies at the origin). The optimal such polynomial reward functions are given by the following optimization problem:

$$
\begin{aligned}
\min_{p} \quad & \mathcal{L}(p(c_0), p(c_1), \ldots, p(c_m)) \\
\text{s.t.} \quad & p(x, y) \text{ polynomial of degree at most } d \\
& p(0, 0) = 0
\end{aligned}
\tag{3}
$$

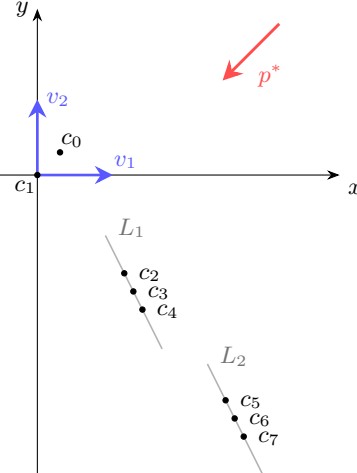

Figure 1: Illustration of the positioning of the candidates for degree $d = 2$. The red arrow labeled $p^*$ indicates the direction of increase of the linear polynomial $p^*(x, y) = -x - y$.

As shown in Claim 2, the polynomial $p^*(x, y) = -x - y$ ranks $c_1$ above $c_0$ when used as a reward function, i.e., $p^*(c_1) > p^*(c_0)$. On the other hand, both voters $v_1$ and $v_2$ rank $c_0$ over $c_1$. In the rest of this proof, our goal will be to show that the optimal polynomial degree-$d$ reward function for our instance (i.e., any solution to (3)) is close to $p^*$, and thus also ranks $c_1$ above $c_0$. This will immediately show that Pareto optimality does not hold for our instance.

**Claim 3.** *In the degenerate instance where $\delta = 0$, and thus $c_0 = c_1 = (0,0)$, the polynomial $p^*(x, y) = -x - y$ is the unique optimal solution of (3).*

*Proof.* By Claim 2, the polynomial $p^*$ achieves the optimal rewards $r^*$ from (2), and thus $p^*$ is an optimal solution of (3) in the degenerate instance where $\delta = 0$. In order to show that $p^*$ is the unique optimal solution, it suffices to show that no other degree-$d$ polynomial achieves the optimal rewards $r^*$, which are the unique optimal solution of (2) by Claim 1. This follows from the fact that the zero polynomial is the only degree-$d$ polynomial that simultaneously vanishes at all points $c_i$, a fact which we prove next.

Consider a polynomial $p$ of degree at most $d$ such that $p(c_i) = 0$ for all $i \in [m]$. We will show that $p = 0$. First, we apply a rotation around the origin to the $(x, y)$ plane such that the lines $L_j$ are now of the form $L_j = \{(x, y) \in \mathbb{R}^2 : y = s_j\}$ for some $0 < s_1 < s_2 < \cdots < s_d$. Note that $p$ is still a polynomial of degree at most $d$ in this new basis.

Now since $p$ vanishes at $d+1$ distinct points on $L_j$, it follows that $p$ restricted to $y = s_j$ is the zero polynomial, i.e., $p\big|_{y=s_j} = 0$. As a result,[4] $p$ must contain a factor $(y-s_j)$ for each $j \in [d]$. Since $p$ has degree at most $d$, it follows that $p$ can be written as $p(x, y) = C \cdot \prod_{j \in [d]} (y - s_j)$. But now $p(0, 0) = r_1^* = 0$, together with $s_j \neq 0$ for all $j$, implies that $C = 0$ and thus $p = 0$. $\square$

We will use Berge's maximum theorem to argue that for sufficiently small $\delta > 0$, any optimal solution to (3) must be close to $p^*$.

**Theorem 4.2** (Berge's Maximum Theorem (Berge, 1997); simplified version). *Let $A \subseteq \mathbb{R}^n$ and $B \subseteq \mathbb{R}^m$ such that $B$ is nonempty and compact. Let $f : A \times B \to \mathbb{R}$ be continuous.*

---

[4]More formally, divide polynomial $p$ by polynomial $y - s_1$ in $(\mathbb{R}[x])[y]$. We obtain $p(x, y) = (y - s_1)q(x, y) + r(x)$, where we note that $r$ only depends on $x$ since it must have degree strictly less than one in $y$. Now $p\big|_{y=s_1} = 0$ implies that $r = 0$. Then, we can continue by dividing $q$ by $y - s_2$ in the same manner to obtain the full factorization.

*Define the set-valued function* $f^* \colon A \rightrightarrows B$ *by* $f^*(a) = \arg\max_{b \in B} f(a, b)$. *Then* $f^*$ *is upper-hemicontinuous with nonempty and compact values.*

**Claim 4.** *There exists a sufficiently small* $\delta > 0$ *such that any optimal solution of (3) satisfies* $p(c_1) > p(c_0)$.

Before proving this claim, let us see why this implies Theorem 4.1. For $\delta$ as given by the claim, any optimal solution ranks $c_1$ over $c_0$. On the other hand, both voters $v_1$ and $v_2$ rank $c_0$ over $c_1$, so Pareto optimality is not satisfied. Furthermore, voter $v_1$ always (trivially) agrees with a strict majority of the voters; in particular, the ordering of voter $v_1$ is PMC for the instance. However, any optimal solution ranks $c_1$ over $c_0$, even though a majority (in fact, everyone) agrees to the opposite. Thus, the example also fails PMC.

*Proof.* In order to use Berge's maximum theorem, we need to make sure that the domain over which we are optimizing is compact. Let $S = \{s = (s_{i,j})_{i,j \geq 0 : i+j \leq d} : s_{i,j} \in \mathbb{R}, s_{0,0} = 0\}$ represent all polynomials of degree at most $d$ with zero constant coefficient, i.e., for each $s \in S$, the corresponding polynomial is given by $p_s(x, y) = \sum_{i=0}^{d} \sum_{j=0}^{d-i} s_{i,j} x^i y^j$. Let $S'$ denote the subset of $S$ where all coefficients have magnitude at most 2, i.e., $S' = \{s \in S : |s_{i,j}| \leq 2\}$. Let $s^* \in S$ be such that $p_{s^*} = p^*$, i.e., $s_{1,0}^* = s_{0,1}^* = -1$ and all other coefficients are zero. Note that $S'$ is compact and $s^*$ lies in the interior of $S'$.

Now define $f : [0, 1] \times S' \to \mathbb{R}$, $(\delta, s) \mapsto \mathcal{L}(p_s(c_0(\delta)), p_s(c_1), \ldots, p_s(c_m))$. Clearly, $f$ is continuous, since the loss function $\ell$ is continuous, and $S'$ is nonempty and compact. Thus, by Berge's maximum[5] theorem (Theorem 4.2), the set-valued function $f^* : [0, 1] \rightrightarrows S', \delta \mapsto \arg\min_{s \in S'} f(\delta, s)$ is upper-hemicontinuous.

Recall that by Claim 3, when $\delta = 0$, the polynomial $p^*(x, y) = -x - y$ is the unique minimizer of $\mathcal{L}$, i.e., $f^*(0) = \{s^*\}$, where $s^* \in S$ is as defined above such that $p^* = p_{s^*}$. As shown in Claim 2, the polynomial $p^* = p_{s^*}$ satisfies $p^*(c_1) > p^*(c_0)$. By continuity, it follows that there exists small enough $\varepsilon \in (0, 1)$, such that we also have $p_s(c_1) > p_s(c_0)$ for all $s \in S$ with $\|s - s^*\|_\infty \leq \varepsilon$. Now, since $f^*$ is upper-hemicontinuous, we know that for sufficiently small $\delta \in (0, 1]$, we have $\|s - s^*\|_\infty \leq \varepsilon$ for all $s \in f^*(\delta)$. In particular, since $\varepsilon < 1$, $f^*(\delta)$ lies in the interior of $S'$. By convexity of the function $s \mapsto \mathcal{L}(p_s(c_0(\delta)), p_s(c_1), \ldots, p_s(c_m))$, it follows that $f^*(\delta)$ is the set of minimizers over all of $S$ (not just $S'$). So for sufficiently small $\delta > 0$, any minimizer $p$ of (3) must satisfy $p(c_1) > p(c_0)$. $\qquad \square$

## 5 RECOVERING PARETO OPTIMALITY WITH UNIFORM DATA

In this section, we return to the setting of linear social choice. We consider an idealized setting where (i) the dataset contains comparisons in all possible directions of space, (ii) each of these comparisons has unit length, and (iii) each direction appears uniformly. More formally, this corresponds to a setting where the total loss can be written as

$$\mathcal{L}(\theta) := \sum_{i=1}^{n} \int_{x \in S^{d-1}, \langle v_i, x \rangle \geq 0} \ell(-\langle \theta, x \rangle) \, dx \tag{4}$$

where $v_1, \ldots, v_n \in \mathbb{R}^d \setminus \{0\}$ are the voters (inducing a ranking according to the corresponding reward function $r_{v_i}(x) = \langle v_i, x \rangle$).

We can define a version of PO over this "complete" dataset.

**Definition 3.** *We say that* $\theta \in \mathbb{R}^d$ *is Pareto optimal (PO) over* $\mathbb{R}^d$ *with respect to voters* $v_1, \ldots, v_n \in \mathbb{R}^d \setminus \{0\}$ *if whenever a direction* $x \in S^{d-1}$ *satisfies* $\langle v_i, x \rangle > 0$ *for all* $i$, *we also have* $\langle \theta, x \rangle > 0$.

**Theorem 5.1.** *In the idealized setting, any loss-based voting rule with a loss function* $\ell$ *that is convex, nondecreasing, lower bounded, and differentiable with* $\ell'(0) > 0$, *satisfies PO, as long as there are at least two distinct voters.*

---

[5]The theorem clearly also applies to minimization by replacing $f$ by $-f$.

On the other hand, we can show that the analogous definition of Pairwise Majority Consistency (PMC) is not satisfied even in this setting.

**Definition 4.** We say that $\theta \in \mathbb{R}^d$ is pairwise majority consistent (PMC) over $\mathbb{R}^d$ with respect to voters $v_1, \ldots, v_n \in \mathbb{R}^d \setminus \{0\}$ if whenever a direction $x \in S^{d-1}$ satisfies $\langle v_i, x \rangle > 0$ for a strict majority of all $i$, we also have $\langle \theta, x \rangle > 0$. A voting rule satisfies PMC if it outputs a PMC vector whenever one exists.

**Theorem 5.2 (♦).** *In the idealized setting, any loss-based voting rule with a loss function $\ell$ that is strictly convex, nondecreasing, lower bounded, and differentiable with $\ell'(0) > 0$ fails PMC.*

Although PMC fails in this uniform data setting, it is not at all obvious whether PMC is even desirable here. For example, if a $p$ fraction of votes come from direction $v_1$ and a $(1-p)$ fraction from direction $v_2$, then in the uniform data setting PMC becomes a discontinuous requirement: the rule must output direction $v_1$ whenever $p > 0.5$ and direction $v_2$ whenever $p < 0.5$. In many applications, one may instead prefer to interpolate between the two. See the work by Lederer et al. (2024) for a recent discussion of similar considerations in rank aggregation (from which PMC is inherited).

## 5.1 Proof of Theorem 5.1

Let $\mathcal{L}_i$ denote the loss with respect to voter $i$, i.e.,

$$\mathcal{L}_i(\theta) := \int_{x \in S^{d-1}, \langle v_i, x \rangle \geq 0} \ell(-\langle \theta, x \rangle)\, dx = \frac{1}{2} \int_{x \in S^{d-1}} \ell(-\operatorname{sgn}(\langle v_i, x \rangle) \cdot \langle \theta, x \rangle)\, dx,$$

where the sign function is defined as

$$\operatorname{sgn}(t) = 1 \text{ if } t > 0, \quad \operatorname{sgn}(t) = 0 \text{ if } t = 0, \quad \operatorname{sgn}(t) = -1 \text{ if } t < 0.$$

Then we can write $\mathcal{L}(\theta) = \sum_{i=1}^{n} \mathcal{L}_i(\theta)$.

**Claim 5 (♦).** *If $\theta, \theta' \in \mathbb{R}^d$ satisfy $\|\theta\|_2 = \|\theta'\|_2$ and $\langle \theta', v_i \rangle > \langle \theta, v_i \rangle$, then $\mathcal{L}_i(\theta') < \mathcal{L}_i(\theta)$.*

**Claim 6 (♦).** *If there are at least two distinct voters, then $\mathcal{L}$ attains its minimum.*

**Claim 7 (♦).** *If there exists $x \in S^{d-1}$ such that $\langle v_i, x \rangle > 0$ for all voters $i \in [n]$, then $\theta = 0$ is not a minimum of $\mathcal{L}$.*

Finally, we use these three claims to prove the following.

**Claim 8.** *Any minimizer $\theta^*$ of $\mathcal{L}$ satisfies PO over $\mathbb{R}^d$ with respect to the voters $v_1, \ldots, v_n$.*

*Proof.* First of all, note that if there does not exist $x \in S^{d-1}$ such that $\langle v_i, x \rangle > 0$ for all $i \in [n]$, then PO is trivially satisfied. Thus, from now on assume that the set $D := \{x \in S^{d-1} : \langle v_i, x \rangle > 0 \text{ for all } i \in [n]\}$ is not empty. In particular, by Claim 7, this implies any minimizer $\theta^*$ of $\mathcal{L}$ is not the zero vector.

Consider any $\theta \neq 0$ such that there exists $x \in D$ such that $\langle \theta, x \rangle \leq 0$. We will show that there exists $\theta'$ with $\mathcal{L}(\theta') < \mathcal{L}(\theta)$ and thus $\theta$ is not optimal. Since the set $D$ is open, we can assume without loss of generality that in fact $\langle \theta, x \rangle < 0$. Furthermore, for simplicity we will assume that $\|\theta\|_2 = 1 = \|x\|_2$. If that is not the case, then one can simply scale $x$ so that it has the same length as $\theta$ and the same proof idea applies.

Let $\theta' := (1 + \delta)\theta + \varepsilon x$. We will show that for some carefully selected $\varepsilon, \delta > 0$, $\theta'$ satisfies $\langle \theta', v_i \rangle > \langle \theta, v_i \rangle$ and $\|\theta'\|_2 = \|\theta\|_2 = 1$. As a result, using Claim 5, it will follow that $\mathcal{L}_i(\theta') < \mathcal{L}_i(\theta)$ for all $i \in [n]$, and thus $\mathcal{L}(\theta') < \mathcal{L}(\theta)$, as desired.

The condition $\langle \theta', v_i \rangle > \langle \theta, v_i \rangle$ holds as long as $\delta \langle v_i, \theta \rangle + \varepsilon \langle v_i, x \rangle > 0$, which holds as long as $\varepsilon > -\frac{\langle v_i, \theta \rangle}{\langle v_i, x \rangle} \delta$ holds for all $i \in [n]$. Let $M := 2 + \max\{0, -1/\langle \theta, x \rangle, \max_i -\langle v_i, \theta \rangle / \langle v_i, x \rangle\}$. Then letting $\varepsilon := M\delta$ ensures that the aforementioned condition is satisfied.

It remains to pick $\delta > 0$ such that $\|\theta'\|_2 = 1$. We can write

$$
\begin{aligned}
\|\theta'\|_2^2 = \langle \theta', \theta' \rangle &= \langle (1+\delta)\theta + M\delta x, (1+\delta)\theta + M\delta x \rangle \\
&= (1+\delta)^2 + 2(1+\delta)M\delta\langle\theta, x\rangle + M^2\delta^2 \\
&= 1 + 2\delta + \delta^2 + 2M\delta\langle\theta, x\rangle + 2\delta^2 M\langle\theta, x\rangle + M^2\delta^2 \\
&= 1 + \delta^2(1 + 2M\langle\theta, x\rangle + M^2) + \delta(2 + 2M\langle\theta, x\rangle).
\end{aligned}
$$

Note that $1 + 2M\langle\theta, x\rangle + M^2 \geq 1 - 2M + M^2 > 0$ and $2 + 2M\langle\theta, x\rangle < 0$, by the choice of $M$ and since $\langle\theta, x\rangle \in [-1, 0)$. As a result, we have that for sufficiently large $\delta > 0$, $\|\theta'\|_2 > 1$, and for sufficiently small $\delta > 0$, $\|\theta'\|_2 < 1$. By the intermediate value theorem, if follows that there exists $\delta > 0$ such that $\|\theta'\|_2 = 1$, and this completes the proof. □

Theorem 5.1 follows by combining Claim 6, which guarantees that a minimizer exists, with Claim 8, which guarantees that any minimizer satisfies PO.

## 6 Discussion and Future Directions

We contribute to the recent effort to import ideas from social choice—especially the axiomatic approach—into the study of AI alignment, pluralistic alignment and multi-objective alignment. At the same time, classical social choice is largely framed for discrete settings (e.g., complete rankings over a fixed option set), which do not directly reflect modern ML pipelines. If social choice is to inform alignment, the axioms and tools must be adapted to the knobs that actually exist in these pipelines.

In this spirit, we analyze why a seemingly minimal axiom such as Pareto Optimality (PO) can fail. Seeking models that better reflect training constraints, we focus on loss-based rules and pursue theoretically tractable proxies for practice: (1) enlarging the reward class, (2) requiring guarantees that hold out-of-sample, and (3) making the data-generating distribution explicit. The first point acknowledges that, while many analyses assume unconstrained rewards, such optima are unlikely to be realized in practice. For (1), we establish a negative result for bounded-degree polynomial rewards and conjecture that fixed-width, fixed-depth MLPs over a frozen embedding space fail for similar reasons.

As a step toward generalization, we show that in structured settings (e.g., linear social choice), axioms specified over a fixed alternative set admit natural analogues over the entire embedding space. In particular, in the linear social choice model, pairwise judgments between embeddings $a$ and $b$ can inform different queries such as $a + \varepsilon$ vs. $b - \varepsilon$ (noisy perturbations) or $a + r$ vs. $b + r$ (shared direction, translated), providing leverage beyond the observed preferences. For (3), we prove that PO is achievable under data distributions in which both inter-embedding distances and directions are suitably uniform/balanced. Taking our simplified model as given, one can then examine the distribution of embedding differences—e.g., via PCA—to assess how close it is to this regime.

While one can balance the lengths of comparison directions by renormalizing the loss, achieving more uniform directional coverage may require reweighting comparisons, and in the worst case, collecting additional data. A natural next step is to derive explicit sampling bounds in the uniform setting. Beyond random sampling, a promising direction is to formalize reweighting and data-query strategies—can we gather data to attain PO in a more targeted (and more efficient) manner?

While proposing new axioms is not our focus, future research should consider how other axioms extend to embedding spaces and can be satisfied through data selection. Overall, our results point to a different way of addressing axiomatic violations within loss-based frameworks.

## Acknowledgments

Sonja Kraiczy was supported by the PIBBSS Fellowship (Principles of Intelligence) and the Cooperative AI Foundation.

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

## A Additional Related Work

As an alternative to unguided preference elicitation ("which response do you prefer?"), several authors (Bai et al., 2022a; Glaese et al., 2022) proposed principle-guided supervision: annotators judge responses with respect to explicit criteria such as helpfulness, honesty, and harmlessness (HHH). In current practice, RLHF often elicits binary preferences from users as a general signal without specifying how to evaluate "better," and human comparisons are noisy—commonly modeled with the Bradley–Terry framework. Asking raters to compare responses with respect to a named principle (e.g., "which is more harmful?") provides a clearer target and greater control, reducing variance from idiosyncratic tastes (e.g., preferring a flattering style). Taking this perspective, we can view the principles as the voters. As models become more capable, human raters can be replaced or augmented with model-based judges, yielding Reinforcement Learning from AI Feedback (RLAIF). This makes alignment methods more scalable. E.g., it is cheap to score the same pairwise comparison against multiple principles, which may legitimately disagree (a response can be more helpful yet more harmful). Constitutional AI (CAI) (Bai et al., 2022b) is a concrete framework built around RLAIF: it introduces an explicit "constitution"—a curated set of principles— which judges which of two responses better adheres to a given principle. Notably, when CAI was introduced, Anthropic reported a Pareto improvement on the helpfulness–harmlessness frontier: increased harmlessness at fixed helpfulness relative to RLHF baselines. Many other works implement constitutional-style alignment, often under different labels, by conditioning binary comparisons on explicit criteria (rubrics/attributes), using verifier- or judge-guided feedback, or training separate objectives that are later combined (Cui et al.; Sun et al., 2023; Glaese et al., 2022).

## B Missing proofs from Section 4

### B.1 Proof of Claim 1

*Proof.* Recall that $r_1 = 0$. For each $j \in \{2, \ldots, m\}$, we have that both the terms $(1-\alpha) \cdot \ell(r_j)$ and $\alpha \cdot \ell(-r_j)$ appear in $\mathcal{L}$. Since $\ell$ is convex and $\ell'(0) > 0$, it follows that $\lim_{x \to +\infty} \ell(x) = +\infty$. Furthermore, recall that $\ell$ is lower bounded over $\mathbb{R}$. As a result, it follows that $\mathcal{L}(r) \to +\infty$ when $|r_j| \to +\infty$.

Since the loss function $\ell$ is strictly convex, the function $\phi : (r_2, \ldots, r_m) \mapsto \mathcal{L}(0, 0, r_2, \ldots, r_m)$ is also strictly convex.[6] Thus, together with the previous paragraph, it follows that $\phi$ attains its unique minimum $r^*$.

It remains to argue that the optimal rewards are ordered $0 = r_1^* < r_2^* < \cdots < r_m^*$. To prove this, we first introduce some additional notation. Define $h : \mathbb{R} \to \mathbb{R}$, $x \mapsto \alpha \cdot \ell(x) + (1-\alpha) \cdot \ell(-x)$. Note that $h$ is convex and differentiable, and thus its derivative $h'$ is nondecreasing. Furthermore, it satisfies

$$h'(0) = \alpha \cdot \ell'(0) - (1-\alpha) \cdot \ell'(0) = (2\alpha - 1) \cdot \ell'(0) > 0$$

since $\alpha > 1/2$ and $\ell'(0)$. We can rewrite the total loss function as

$$\mathcal{L}(0, 0, r_2, \ldots, r_m) = \ell(0) + 2 \sum_{j=2}^{m} h(-r_j) + \sum_{i=2}^{m-1} \sum_{j=i+1}^{m} h(r_i - r_j)$$

and thus the partial derivatives for all $i \geq 2$ as

$$\frac{\partial \mathcal{L}}{\partial r_i}(r) = -2h'(-r_i) - \sum_{k=2}^{i-1} h'(r_k - r_i) + \sum_{k=i+1}^{m} h'(r_i - r_k).$$

Assume towards a contradiction that $r_i^* \geq r_{i+1}^*$ for some $i \in \{2, \ldots, m-1\}$. We will handle the remaining case $i = 1$ separately at the end. Since $r^*$ is the optimal solution and $\mathcal{L}$ is

---

[6]Here we use the fact that for $(r_2, \ldots, r_m) \neq (r'_2, \ldots, r'_m)$, there exists $j \in \{2, \ldots, m\}$ such that $r_j \neq r'_j$, and thus $\mathcal{L}$ contains a term for which $\ell((r_j + r'_j)/2) < \ell(r_j)/2 + \ell(r'_j)/2$.

differentiable, we have $\partial\mathcal{L}/\partial r_i(r^*) = 0$ for all $i \geq 2$. However, we can write

$$\frac{\partial\mathcal{L}}{\partial r_{i+1}}(r^*) - \frac{\partial\mathcal{L}}{\partial r_i}(r^*) = 2(h'(-r_i^*) - h'(-r_{i+1}^*)) - h'(r_i^* - r_{i+1}^*) - h'(r_i^* - r_{i+1}^*)$$

$$+ \sum_{k=2}^{i-1}(h'(r_k^* - r_i^*) - h'(r_k^* - r_{i+1}^*)) + \sum_{k=i+2}^{m}(h'(r_{i+1}^* - r_k^*) - h'(r_i^* - r_k^*))$$

$$\leq -2h'(r_i^* - r_{i+1}^*) < 0$$

where in the first inequality we used the fact that $r_i^* \geq r_{i+1}^*$ and $h'$ is nondecreasing. In the second inequality we used $h'(r_i^* - r_{i+1}^*) \geq h'(0) > 0$. Since the partial derivatives are zero, this is a contradiction. So, it must be that $r_i^* < r_{i+1}^*$ for all $i \geq 2$.

It remains to show that $r_1^* < r_2^*$, i.e., $r_2^* > 0$. Assume towards a contradiction that $r_2^* \leq 0$. We can write

$$\sum_{i=2}^{m}\frac{\partial\mathcal{L}}{\partial r_i}(r^*) = -2\sum_{i=2}^{m}h'(-r_i^*) + \sum_{i=2}^{m-1}\sum_{j=i+1}^{m}(h'(r_i^* - r_j^*) - h'(r_i^* - r_j^*)) = -2\sum_{i=2}^{m}h'(-r_i^*)$$

since every term $h'(r_i^* - r_j^*)$ appears once with a positive sign and once with a negative sign. Thus, we can write

$$\sum_{i=2}^{m}\frac{\partial\mathcal{L}}{\partial r_i}(r^*) + 2\frac{\partial\mathcal{L}}{\partial r_2}(r^*) = -2\sum_{i=2}^{m}h'(-r_i^*) - 4h'(-r_2^*) + 2\sum_{k=3}^{m}h'(r_2^* - r_k^*)$$

$$= -6h'(-r_2^*) + 2\sum_{k=3}^{m}(h'(r_2^* - r_k^*) - h'(-r_k^*))$$

$$\leq -6h'(-r_2^*) < 0$$

where in the first inequality we used the fact that $r_2^* \leq 0$ and $h'$ is nondecreasing. In the second inequality we used $h'(-r_2^*) \geq h'(0) > 0$. Since all the partial derivatives are zero, this is a contradiction. So we also have $r_2^* > 0 = r_1^*$. $\qquad\square$

## C  Missing proofs from Section 5

### C.1  Proof of Claim 5

*Proof.* First of all, note that since $\|\theta\|_2 = \|\theta'\|_2$ and $\langle\theta', v_i\rangle > \langle\theta, v_i\rangle$, it must be that $\theta \neq 0$ and $\theta' \neq 0$. Without loss of generality, since the statement of the claim does not change if we rotate the space, we can assume that $v_i = e_1$ is the unit length vector with entry 1 in the first dimension and 0 otherwise.

Next, we argue that, without loss of generality, we can assume that $\theta_2 \geq 0$ and $\theta_j = 0$ for all $j \geq 3$, and similarly for $\theta'$. Let $R' : \mathbb{R}^{d-1} \to \mathbb{R}^{d-1}$ denote the rotation around the origin in $(d-1)$-dimensional space that maps $(\theta_2, \ldots, \theta_d)$ to $(\alpha, 0, \ldots, 0)$, where $\alpha \geq 0$. Define $R : \mathbb{R}^d \to \mathbb{R}^d$ by $R(x) = (x_1, R'(x_2, \ldots, x_d))$. Note that $\|R(\theta)\|_2 = \|\theta\|_2$ and $\langle R(\theta), v_i\rangle = \langle\theta, v_i\rangle$, since $v_i = e_1$. We will show that $\mathcal{L}_i(R(\theta)) = \mathcal{L}_i(\theta)$. Note that $R$ is a bijection and we have

$$\langle R(\theta), x\rangle = \theta_1 x_1 + \langle R'(\theta_2, \ldots, \theta_d), (x_2, \ldots, x_d)\rangle = \theta_1 x_1 + \langle(\theta_2, \ldots, \theta_d), R'^{-1}(x_2, \ldots, x_d)\rangle$$

$$= \langle\theta, R^{-1}(x)\rangle$$

where $R'^{-1}$ is the inverse rotation to $R'$. We can thus write

$$\mathcal{L}_i(R(\theta)) = \int_{x \in S^{d-1}, x_1 \geq 0}\ell(-\langle R(\theta), x\rangle)\,dx = \int_{x \in S^{d-1}, x_1 \geq 0}\ell(-\langle\theta, R^{-1}(x)\rangle)\,dx$$

$$= \int_{x \in S^{d-1}, x_1 \geq 0}\ell(-\langle\theta, x\rangle)\,dx$$

$$= \mathcal{L}_i(\theta)$$

since $R^{-1}$ is a smooth bijection that preserves distances. Thus, we can assume that $\theta_2 \geq 0$ and $\theta_j = 0$ for all $j \geq 3$, and similarly for $\theta'$.

Let $T$ be the rotation that maps $\theta$ to $\theta'$. We can write

$$
\begin{aligned}
\mathcal{L}(\theta) &= \int_{x \in S^{d-1}, x_1 \geq 0} \ell(-\langle \theta, x \rangle)\, dx \\
&= \int_{x \in S^{d-1}, [T^{-1}(x)]_1 \geq 0} \ell(-\langle \theta, T^{-1}(x) \rangle)\, dx \\
&= \int_{x \in S^{d-1}, [T^{-1}(x)]_1 \geq 0} \ell(-\langle T(\theta), x \rangle)\, dx \\
&= \int_{x \in S^{d-1}, [T^{-1}(x)]_1 \geq 0, x_1 \geq 0} \ell(-\langle \theta', x \rangle)\, dx + \int_{x \in S^{d-1}, [T^{-1}(x)]_1 \geq 0, x_1 \leq 0} \ell(-\langle \theta', x \rangle)\, dx.
\end{aligned}
$$

We can also decompose

$$
\begin{aligned}
\mathcal{L}(\theta') &= \int_{x \in S^{d-1}, x_1 \geq 0} \ell(-\langle \theta', x \rangle)\, dx \\
&= \int_{x \in S^{d-1}, [T^{-1}(x)]_1 \geq 0, x_1 \geq 0} \ell(-\langle \theta', x \rangle)\, dx + \int_{x \in S^{d-1}, [T^{-1}(x)]_1 \leq 0, x_1 \geq 0} \ell(-\langle \theta', x \rangle)\, dx.
\end{aligned}
$$

Thus, we obtain

$$
\begin{aligned}
\mathcal{L}(\theta) - \mathcal{L}(\theta') &= \int_{x \in S^{d-1}, [T^{-1}(x)]_1 \geq 0, x_1 \leq 0} \ell(-\langle \theta', x \rangle)\, dx - \int_{x \in S^{d-1}, [T^{-1}(x)]_1 \leq 0, x_1 \geq 0} \ell(-\langle \theta', x \rangle)\, dx \\
&= \int_{x \in S^{d-1}, [T^{-1}(x)]_1 \leq 0, x_1 \geq 0} \ell(\langle \theta', x \rangle) - \ell(-\langle \theta', x \rangle)\, dx \\
&= \int_{x \in S^{d-1}, [T^{-1}(x)]_1 < 0, x_1 > 0} \ell(\langle \theta', x \rangle) - \ell(-\langle \theta', x \rangle)\, dx.
\end{aligned}
$$

Note that the domain $\{x \in S^{d-1}, [T^{-1}(x)]_1 < 0, x_1 > 0\}$ has positive measure, since $\langle \theta', v_i \rangle > \langle \theta, v_i \rangle$. Furthermore, we have $\ell(t) > \ell(-t)$ for all $t > 0$.[7] As a result, if we can show that $\langle \theta', x \rangle > 0$ for all $x \in S^{d-1}$ with $[T^{-1}(x)]_1 < 0$ and $x_1 > 0$, then we will obtain $\mathcal{L}(\theta) - \mathcal{L}(\theta') > 0$, as desired.

Recall that we assume without loss of generality that $\theta_2, \theta_2' \geq 0$ and $\theta_j = \theta_j' = 0$ for all $j \geq 3$. Furthermore, by assumption we have $\theta_1' > \theta_1$. Consider any $x \in S^{d-1}$ with $\langle \theta', x \rangle \leq 0$ and $x_1 > 0$. Assume towards a contradiction that $[T^{-1}(x)]_1 < 0$. The rotation $T^{-1}$ maps $\theta'$ to $\theta$, which both lie in the upper hemisphere, and $\theta_1' > \theta_1$. Thus, the rotation is counter-clockwise and of angle at most $\pi$. Since $x_1 > 0$, we can assume without loss of generality that $\theta_1 < 0$ and $\theta_2 = 0$, i.e., $\theta$ is in the negative $x$ axis direction. Indeed, since $x_1 > 0$, if rotating $x$ in the counterclockwise direction by some angle $\alpha < \pi$ yields a point $y$ with $y_1 < 0$, then rotating it by any angle $\beta \in (\alpha, \pi]$ will also yield a point $y$ with $y_1 < 0$. Now it is easy to see that the rotation $T^{-1}$ that maps $\theta'$ to the negative axis direction also maps any point $x$ with $\langle \theta', x \rangle \leq 0$ to a point $y = T^{-1}(x)$ with $y_1 \geq 0$, a contradiction. $\qquad\square$

## C.2 Proof of Claim 6

*Proof.* We show that for any $M \in \mathbb{R}$, there exists a $t > 0$ such that if $\|\theta\|_2 > t$ then $\mathcal{L}(\theta) > M$. Then, since $\mathcal{L}(0)$ is finite, we can conclude that $\mathcal{L}$ attains its minimum. Observe that since at least two voters are distinct, say $v_1$ and $v_2$, there exists $\kappa < 1$ such that for any $\theta \in S^{d-1}$, there exists $i \in \{1, 2\}$ such that $\langle \theta, v_i \rangle \leq \kappa$. Here we assume that $\|v_1\|_2 = \|v_2\|_2 = 1$ without loss of generality.

Let $M$ be arbitrary. Let $\theta \in S^{d-1}$ be arbitrary. Without loss of generality, say that voter $v_1 = e_1$ is such that $\langle \theta, v_1 \rangle \leq \kappa$. Since the loss function $\ell$ is lower bounded, there exists

---

[7]This follows from the fact that $\ell$ is nondecreasing and $\ell'(0) > 0$.

$K < 0$ such that

$$\mathcal{L}(t\theta) \geq K + \mathcal{L}_1(t\theta) \geq K + \int_{x \in S^{d-1}, x_1 \geq 0} \ell(-\langle t\theta, x \rangle) \, dx$$

$$\geq 2K + \int_{x \in S^{d-1}, x_1 \geq 0, \langle \theta, x \rangle \leq f(\kappa, d)} \ell(-\langle t\theta, x \rangle) \, dx$$

where $f(\kappa, d) < 0$ is such that the set $\{x \in S^{d-1} : x_1 \geq 0, \langle \theta, x \rangle \leq f(\kappa, d)\}$ has strictly positive measure bounded away from zero by at least $g(\kappa, d) > 0$. As a result, we obtain that

$$\mathcal{L}(t\theta) \geq -2K + g(\kappa, d) \cdot \ell(-t \cdot f(\kappa, d)) > M$$

for a sufficiently large $t > 0$, since $\ell$ is strictly increasing for positive inputs.[8] $\qquad \square$

### C.3  Proof of Claim 7

*Proof.* Let $z \in S^{d-1}$ be such that $\langle v_i, z \rangle > 0$ for all voters $i \in [n]$. We can write

$$\mathcal{L}_i(t \cdot z) = \int_{x \in S^{d-1}, \langle v_i, x \rangle \geq 0} \ell(-\langle t \cdot z, x \rangle) \, dx.$$

Taking the derivative with respect to $t$ at $t = 0$ we obtain

$$\frac{d}{dt}\Big|_{t=0} \mathcal{L}_i(t \cdot z) = \int_{x \in S^{d-1}, \langle v_i, x \rangle \geq 0} -\langle z, x \rangle \ell'(0) \, dx = \ell'(0) \int_{x \in S^{d-1}, \langle v_i, x \rangle \geq 0} -\langle z, x \rangle \, dx < 0$$

which follows from $\ell'(0) > 0$ and

$$\int_{x \in S^{d-1}, \langle v_i, x \rangle \geq 0} \langle z, x \rangle \, dx > 0$$

which follows from $\langle v_i, z \rangle > 0$ using similar ideas to those in the proof of Claim 5. $\qquad \square$

### C.4  Proof of Theorem 5.2

*Proof.* Consider $\mathbb{R}^2$ and let $v_1 = (0, 1)$, $v_2 = (1, 0)$. For $p \in [1/4, 1/2]$, we consider the instance where a fraction $p$ of the voters are in direction $v_1$, and the remaining fraction $(1 - p)$ are in direction $v_2$. Thus, the total loss function that is minimized over $\theta \in \mathbb{R}^2$ can equivalently be written as

$$\mathcal{L}(\theta, p) := 2p \int_{x \in S^1, x_2 \geq 0} \ell(-\langle \theta, x \rangle) \, dx + 2(1 - p) \int_{x \in S^1, x_1 \geq 0} \ell(-\langle \theta, x \rangle) \, dx.$$

By Claim 6, $\mathcal{L}(\cdot, p)$ attains its minimum for each $p \in [1/4, 1/2]$. Furthermore, since $\ell$ is strictly convex, so is[9] $\mathcal{L}(\cdot, p)$, and thus the minimizer of $\mathcal{L}(\cdot, p)$ is unique.

For any $p < 1/2$, $v_2$ is the strict majority voter and so $\theta = v_2$ is the only pairwise majority consistent direction. In order to show that PMC is not satisfied, it thus suffices to prove that for some $p < 1/2$, the minimizer of the total loss is different from direction $v_2$.

First, for $p = 1/2$ we can write

$$\mathcal{L}(\theta, 1/2) = \int_{x \in S^1, x_2 \geq 0} \ell(-\langle \theta, x \rangle) \, dx + \int_{x \in S^1, x_1 \geq 0} \ell(-\langle \theta, x \rangle) \, dx$$

$$= \int_{x \in S^1, x_1 \geq 0} \big( \ell(-\langle R(\theta), x \rangle) + \ell(-\langle \theta, x \rangle) \big) \, dx,$$

---

[8]This follows from the fact that $\ell$ is convex and $\ell'(0) > 0$.

[9]This follows from standard results in convex analysis (Rockafellar, 1970) together with the observation that for any $a \neq b$, the set of points $x$ on $S^1$ such that $\langle a - b, x \rangle = 0$ has measure zero (on $S^1$).

where $R$ denotes the 90° clockwise rotation, i.e., $R(\theta) = (\theta_2, -\theta_1)$. Let $\theta' = (\theta_2, \theta_1)$ and note that $\langle R(\theta), x \rangle = \theta_2 x_1 - \theta_1 x_2 = \langle \theta', (x_1, -x_2) \rangle$. Since the domain of integration is symmetric with respect to the transformation $(x_1, x_2) \mapsto (x_1, -x_2)$, we obtain

$$\mathcal{L}(\theta, 1/2) = \int_{x \in S^1, x_1 \geq 0} \left( \ell(-\langle \theta', x \rangle) + \ell(-\langle \theta, x \rangle) \right) dx.$$

It follows that $\mathcal{L}(\theta, 1/2) = \mathcal{L}(\theta', 1/2)$, i.e., the loss is invariant under permuting $\theta_1$ and $\theta_2$. By convexity,

$$\mathcal{L}\left((\theta + \theta')/2, 1/2\right) \leq \frac{1}{2}\left(\mathcal{L}(\theta, 1/2) + \mathcal{L}(\theta', 1/2)\right) = \mathcal{L}(\theta, 1/2),$$

so the unique minimizer $\theta^*$ of $\mathcal{L}(\cdot, 1/2)$ satisfies $\theta_1^* = \theta_2^*$. From Claim 7, we know $\theta^* \neq 0$. By Theorem 5.1, PO holds for this instance, so $\theta^*$ lies in the positive cone of $v_1$ and $v_2$, and in particular $\theta_2^* > 0$.

Now let $M > 3\|\theta^*\|$ and $B(M) = \{\theta \in \mathbb{R}^2 : \|\theta\| \leq M\}$. The map $\mathcal{L}: B(M) \times [1/4, 1/2] \to \mathbb{R}$ is continuous, so by Berge's theorem (Theorem 4.2) the function[10]

$$\mathcal{L}^*: [1/4, 1/2] \to B(M), \qquad \mathcal{L}^*(p) = \underset{\theta \in B(M)}{\arg\min}\, \mathcal{L}(\theta, p)$$

is continuous. Thus, for sufficiently small $\delta \in (0, 1/4)$,

$$\|\mathcal{L}^*(1/2 - \delta) - \theta^*\| < \theta_2^* \leq \|\theta^*\|,$$

so

$$\|\mathcal{L}^*(1/2 - \delta)\| \leq \|\theta^*\| + \|\mathcal{L}^*(1/2 - \delta) - \theta^*\| \leq 2\|\theta^*\| < M.$$

In particular, $\mathcal{L}^*(1/2 - \delta)$ does not lie on the boundary of $B(M)$, and therefore also minimizes $\mathcal{L}(1/2 - \delta, \theta)$ over all $\theta \in \mathbb{R}^2$.

Finally, any $(t, 0)$ with $t > 0$ has distance at least $\theta_2^*$ from $\theta^*$, so $\mathcal{L}^*(1/2 - \delta)$ is not a scalar multiple of the majority voter $v_2 = (1, 0)$, and therefore PMC is not satisfied when $p = 1/2 - \delta$. □

---

[10]For each $p \in [1/4, 1/2]$, the minimizer is unique.

