# OpenReview forum: "Enforcing Axioms for AI Alignment under Loss-Based Rules"
_ICLR.cc/2026/Conference — ICLR 2026 Poster_

### Official Review · Reviewer_6CiF · 2025-10-29

**Soundness:** 3
**Presentation:** 3
**Contribution:** 3
**Rating:** 8
**Confidence:** 3

**Summary:**

This paper studies axiomatic violations in AI alignment, specifically how loss-based reward models can violate Pareto optimality even when all principles agree on preferences. The authors show polynomial rewards don't resolve this issue, but prove that uniform data coverage can restore axiomatic guarantees in constitutional-style alignment methods.

**Strengths:**

1. The paper provides rigorous mathematical analysis of axiomatic violations in AI alignment, extending the linear social choice framework with formal proofs and theorems.
2. The paper proves that even polynomial reward functions (beyond linear) still fail Pareto optimality, strengthening the robustness of the violation result.
3. The paper provides intuitive examples and simplified cases that clearly illustrate why Pareto optimality violations occur

**Weaknesses:**

Only examines Pareto optimality while other important social choice axioms (e.g., Pairwise Majority Consistency) receive limited attention.

**Questions:**

None

---

> ### Author Response · Authors · 2025-11-24
>
> We thank the reviewer for their thoughtful review and positive assessment of our paper.
>
> The only weakness pointed out in the review is the following:
> >Only examines Pareto optimality while other important social choice axioms (e.g., Pairwise Majority Consistency) receive limited attention.
>
> To us, the appeal of PO is that it is a minimal and largely uncontroversial axiom. In this paper, we did not focus on defending (let alone proposing) stronger axioms (such as PMC). Instead, we chose to work towards theoretical approaches such as model expressivity, data assumptions, that model components of practical pipelines, to circumvent negative results.
>
> Nevertheless, we appreciate that it is natural to discuss Pairwise Majority Consistency (PMC), since it was studied by Ge et al. in their paper. Therefore, we have added the following results about PMC in the revised version:
> 1. PMC fails to hold even with polynomial reward functions. (This result is obtained using the same construction that we use for the counter-example to PO.)
> 2. In the uniform data setting, PMC still fails to hold. This is in contrast with our result which shows that PO is satisfied.
>
> Although we show that PMC fails in the uniform data setting, it is not at all obvious whether PMC is desirable here. For example, if a $p$ fraction of votes come from $v_1$ and a $(1-p)$ fraction from $v_2$, then in the uniform data setting PMC becomes a discontinuous requirement: the rule must output direction $v_1$ whenever $p>0.5$ and direction $v_2$ whenever $p<0.5$. In many applications one may instead prefer to interpolate between the two. For a recent discussion of similar considerations in rank aggregation (from which PMC is inherited), we refer the reviewer to [1].
>
>
> [1] P. Lederer, D. Peters, and T. Wąs, “The Squared Kemeny Rule for Averaging Rankings,” in Proceedings of the 25th ACM Conference on Economics and Computation (EC 2024)

---

### Official Review · Reviewer_ZJsy · 2025-10-29

**Soundness:** 3
**Presentation:** 3
**Contribution:** 2
**Rating:** 4
**Confidence:** 3

**Summary:**

The paper analyzes axiomatic guarantees (Pareto Optimality) for preference-based alignment under a loss-minimization view. It (i) shows PO violations persist even when moving beyond linear rewards to bounded-degree polynomial rewards; (ii) gives intuition via minimal counterexamples; and (iii) proves that in an idealized setting with uniform directional coverage of pairwise data, broad classes of loss-based rules do satisfy PO—suggesting a data-centric recipe (balanced coverage) for constitutional-style alignment without changing training pipelines.

**Strengths:**

1.  Crisp negative/positive results. Clear impossibility for polynomial rewards and a clean, sufficient condition (uniform coverage) restoring PO.
2.  Training-pipeline relevance. Positions guarantees within standard loss-based training, aligning with RLHF/RLAIF practice rather than bespoke voting rules.
3.  Intuition + formality. Minimal counterexamples illuminate why PO fails; proofs formalize the phenomenon.
4.  Actionable takeaway. Suggests data-centric interventions (balancing directions/lengths) as a path to axioms in practice

**Weaknesses:**

1.  Idealized coverage assumption. The uniform directional coverage condition is strong; guidance on how much imbalance is tolerable is missing (no finite-sample or robustness bounds).
2.  Empirical gap. No experiments (even synthetic) to quantify coverage diagnostics (e.g., spectrum of pairwise direction differences) or to validate the recipe on real preference data.
3.  Scope of axioms. Focuses on PO; discussion of other axioms (e.g., Pairwise Majority Consistency) is limited to related work, with no analogous guarantees.
4.  Practical measurement. The paper hints at PCA/geometry checks but lacks a concrete coverage diagnostic or reweighting algorithm practitioners can run.

**Questions:**

1.  Coverage diagnostics. How should practitioners measure directional coverage in real datasets (e.g., a concrete statistic or test), and what thresholds approximate the theorem’s regime?
2.  Finite-sample guarantees. Can you provide sample-complexity or robustness bounds showing how PO violation probability decays as coverage improves?
3.  Beyond PO. Do analogous results (negative or positive) hold for other alignment-relevant axioms such as Pairwise Majority Consistency under similar coverage assumptions?
4.  Practical recipe. Can you outline a reweighting/sampling procedure (pseudocode) that moves an arbitrary dataset toward the desired coverage while preserving data efficiency?

---

> ### Author Response · Authors · 2025-11-24
>
> We thank the reviewer for these questions. As noted in the conclusion, we already list these directions as important avenues for future work that fall outside the main scope of the present paper.
>
> Nevertheless, we can partially address them here.
>
> ### Question 1 ###
> > Coverage diagnostics. How should practitioners measure directional coverage in real datasets (e.g., a concrete statistic or test), and what thresholds approximate the theorem’s regime?
>
> Given a dataset $C$ and a set of comparisons $D \subset C \times C$, we embed all items in $C$ and take the signed differences between the embeddings of each pair in $D$. Working in a suitably chosen lower-dimensional embedding space in which the relevant principles are approximately linear (see, e.g., [1], which shows that behaviors are linear in a shared intermediate transformer layer), we then normalize these difference vectors to unit length. This reduces the problem to measuring coverage on the unit sphere. Coverage can be quantified via the covering radius, defined as the supremum over points on the sphere of the minimum distance to any point in our set. In practice, this can be approximated by sampling points uniformly at random from the hypersphere and estimating the covering radius.
>
> ### Question 4 ###
> > Practical recipe. Can you outline a reweighting/sampling procedure (pseudocode) that moves an arbitrary dataset toward the desired coverage while preserving data efficiency?
>
> Our response to the first question already partially answers Question 4: we do not require unit-length comparisons a priori. We can simply normalize each difference vector (i.e., divide by its norm) in the Bradley–Terry loss to obtain an alternative loss we optimize, so that we work with unit-length directions without changing the underlying comparisons.
>
> Further to Question 4, our condition has three components: (i) unit-length difference vectors (handled as above), (ii) coverage of directions on the sphere, and (iii) uniformity of their distribution. For (iii), note that even in the limiting case we can easily have a distribution over data points that induces coverage of all directions but is skewed, i.e., a non-uniform distribution on S^{d-1}. In this case, the contribution of each direction can be reweighted by dividing its loss term by the density at that point. For a finite dataset, there are then two natural strategies: if directional coverage is poor (large covering radius), we can add new data so as to reduce the covering radius as defined above; if coverage is good but the distribution is non-uniform, we can instead apply a weighting scheme analogous to the one in [2]. Intuitively, each comparison $(a, b)$ induces a direction/ point $p$ on $S^{d-1}$ and receives a weight determined as follows: each point on $S^{d-1}$ “delegates” its unit density to the closest point in the finite set of unit-length vectors $C'$. Provided the coverage radius is sufficiently small, this could induce weights that downweight directions in dense regions and upweight directions in sparse regions, thereby correcting for non-uniformity.
>
> ### Question 2 ###
> > Finite-sample guarantees. Can you provide sample-complexity or robustness bounds showing how PO violation probability decays as coverage improves?
>
> This is indeed a very natural next direction to investigate. For the general type of loss functions that we consider, it might be too challenging to obtain good sample-complexity guarantees. Thus, a more promising direction would be to study specific natural loss functions and obtain positive results for sampling there. We leave this technical investigation for future work.
>
> ### Question 3 ###
> > Beyond PO. Do analogous results (negative or positive) hold for other alignment-relevant axioms such as Pairwise Majority Consistency under similar coverage assumptions?
>
> We have added the following results about Pairwise Majority Consistency (PMC) in the revised version:
> 1. PMC fails to hold even with polynomial reward functions. (This result is obtained using the same construction that we use for the counter-example to PO.)
> 2. In the uniform data setting, PMC still fails to hold. This is in contrast with our result which shows that PO is satisfied. (But see our response to reviewer 6CiF about why PMC might not be desirable here.)
>
> [1] N. Rimsky, N. Gabrieli, J. Schulz, M. Tong, E. Hubinger, and A. M. Turner, “Steering Llama 2 via Contrastive Activation Addition,” in Proceedings of the 62nd Annual Meeting of the Association for Computational Linguistics (ACL 2024)
>
> [2] A. D. Procaccia, B. Schiffer, and S. Zhang, “Clone-Robust AI Alignment,” in Proceedings of the 42nd International Conference on Machine Learning (ICML), PMLR, vol. 267, pp. 49 903–49 926, 2025.

---

### Official Review · Reviewer_U5VA · 2025-10-30

**Soundness:** 3
**Presentation:** 3
**Contribution:** 4
**Rating:** 8
**Confidence:** 4

**Summary:**

The paper take the impossibility result from Ge et al. (2024) regarding Pareto optimality of reward model training, and (i) generalize it to impossibility for polynomial hypothesis classes (instead of linear ones); (ii) show that the impossibility dissolves when we have preference samples that covers all directions of the embedding spaces uniformly and that the between-response distance in each sample is constant. By doing so, it argues that (i) the impossibility is real and not solved by simply adding representational capacity, yet (ii) can be resolved in practice via data distribution interventions.

**Strengths:**

- Significance: Ge et al. (2024) is a seminal work on an important topic (pluralistic alignment). The work addresses the most important shortcoming of Ge et al., which is its practical implications for today's model training pipelines. It shows what the data distribution needs to be like for the impossibility result to vanish, and the resulting suggestions (uniformly distributed embedding directions & constant embedding distances for preference pairs) seem highly actionable.
- Soundness: I followed the proofs in the body and spot no errors (except possible typo that I mentioned in the questions section). The "claims"/lemmas involved in the proofs seem generally easy to give proof sketches to, so I did not verify their proofs in the appendix.
- Presentation: The authors went with the simplest structure (first background, then present results in logical succession), and it worked well. The message is also clear.

**Weaknesses:**

- It is a pity, although definitely not a fatal flaw, that the paper gives no empirical validation on language models despite its aim at guiding practice. It would be helpful to see, e.g., comparisons of more uniform vs less uniform sampling methods for preference data collection, in terms of impact on downstream alignment performance against multiple constitutional principles.

**Minor points (don't affect score):**
- Presentation: Illlustrations would make following the proofs easier. For the proof of Thm 4.1, an illustration of the parallel lines + candidate points + voter vectors will be useful (same goes for Sec 3). For the proof of Thm 5.1, an illustration of a half-plane (i.e. $\\{x:\\langle x,\\theta\\rangle>0\\}$) gradually rotating towards the intersection $D$ of the voter half-planes will be useful.
- One unmentioned related work is [1] which found circularity in the hypothesis class's privilege graph as a condition for the impossibility. This may give a high-level reason why the impossibility continues to hold for the polynomial hypothesis class.

[1] Representative Social Choice: From Learning Theory to AI Alignment

**Questions:**

- Potential typo: In the proof of Thm 4.1, on line 298, should it be $j=\\lceil\\frac{i}d\\rceil$ instead of $j=(i-1)\\bmod d$?
- What do you think are direct implications for data collection practices in preference alignment / constitutional AI? Are there ways you can quickly test them?

---

> ### Author Response · Authors · 2025-11-24
>
> We thank the reviewer for their thoughtful review and positive assessment of our paper.
>
> ### Regarding the two minor points mentioned by the reviewer ###
> - Illustrations: We thank the reviewer for this suggestion. We have added a figure for the proof of Thm 4.1; see Figure 1 in the revised version of our paper. Due to space constraints, we have not added a figure for Thm 5.1, but we will consider this for the full preprint of our paper.
> - Related work: We thank the reviewer for pointing out this paper. We have added it to our related work section.
>
> ### Response to the first question: ###
> We thank the reviewer for pointing out this typo. It should say $j = \lceil (i-1)/(d+1) \rceil$.
>
> ### Response to the second question: ###
> In our reply to this question we address both the question:
> > What do you think are direct implications for data collection practices in preference alignment / constitutional AI? Are there ways you can quickly test them?"
>
> as well as the weakness mentioned by the reviewer:
> > It is a pity, although definitely not a fatal flaw, that the paper gives no empirical validation on language models despite its aim at guiding practice. It would be helpful to see, e.g., comparisons of more uniform vs less uniform sampling methods for preference data collection, in terms of impact on downstream alignment performance against multiple constitutional principles.
>
> We agree that obtaining empirical validation of our theoretical insights is the next natural step. We provide some first thoughts,  but believe these ideas deserve a thorough and careful empirical validation, and that this can be achieved in a better way by a standalone paper that focuses on that.
>
> Since our positive result is for linear reward functions, it points towards finding an intermediate layer of an LLM where the different principles are approximately linear and fitting a linear reward model. This is essentially a linear probe (LP) and existing work [1, 2] studies using LP as a first stage before using it to fine-tune (“LP → FT”) and finds settings where this outperforms full fine-tuning in out of distribution generalization.
>
> We can use existing datasets of generated data that align with the various principles to various degrees. To make the connection to practice more explicit, recent work (e.g., [3]) in mechanistic interpretability shows that many distinct "behaviors" such as sycophancy, refusal, corrigibility and more, correspond to approximately linear directions in a common fixed intermediate layer of an LLM (e.g., layer 13/14/15 in LLaMA-7B).
>
> Importantly, the unit length of the comparison directions can easily be enforced by a simple pre-processing step: just normalize the vectors from our sample to have length 1. Thus, the only real assumption in our positive theoretical result is that the samples uniformly cover all directions. Thus, for pairs of principles any dataset that looks like a linear transformation of a cloud of points or rotationally symmetric points (Gaussian, disk, etc) would suffice.
>
> The theoretical result thus suggests normalizing the different comparisons; by default it may be that comparisons in some directions have larger differences than others and this might implicitly weight a principle more strongly. Moreover, by looking at the PC1 and PC2 components (e.g., see the PCA visualizations in the associated code repositories of [3]) for pairs of principles one could check if the dataset roughly forms a cloud. Thus the Anthropic datasets and the generated datasets in [3] provide a good starting point, but we believe that such experiments deserve a thorough and careful empirical validation.
>
> [1] A. Tomihari and I. Sato, “Understanding Linear Probing then Fine-tuning Language Models from NTK Perspective,” in Advances in Neural Information Processing Systems (NeurIPS 2024), 2024.
>
> [2] A. Kumar, A. Raghunathan, R. Jones, T. Ma, and P. Liang, “Fine-tuning can distort pretrained features and underperform out-of-distribution,” in International Conference on Learning Representations (ICLR), 2022.
>
> [3] N. Rimsky, N. Gabrieli, J. Schulz, M. Tong, E. Hubinger, and A. M. Turner, “Steering Llama 2 via Contrastive Activation Addition,” in Proceedings of the 62nd Annual Meeting of the Association for Computational Linguistics (ACL 2024), Volume 1: Long Papers, 2024.

---

### Official Review · Reviewer_KC58 · 2025-11-01

**Soundness:** 2
**Presentation:** 1
**Contribution:** 2
**Rating:** 2
**Confidence:** 4

**Summary:**

This paper investigates when loss-based training rules, such as RLHF, satisfy fundamental social-choice axioms, focusing on Pareto Optimality (PO). The authors work in the linear social-choice model of Ge et al. (2024). They extend the analysis to from linear to polynomial reward functions and prove that PO violations persist for any polynomial, establishing a general impossibility. Finally, they take a data-centric perspective: when pairwise-comparison data uniformly cover the embedding space, they prove that PO is satisfied. This yields a practical takeaway, axiomatic guarantees can be restored not by changing loss functions but by ensuring balanced sampling or reweighting of preference data.

**Strengths:**

The main strength of this paper lies in its two key theoretical results.

Theorem 4.1 extends the findings of Ge et al. (2024) from linear to polynomial reward functions, showing that Pareto Optimality (PO) remains violated even when polynomial reward models are used.

Theorem 5.1 demonstrates that under idealized conditions, specifically, unit-length representations and uniformly distributed directions, a linear reward model can in fact satisfy PO.

**Weaknesses:**

The main limitation of this paper lies in its significance comparing to previous results and its practical implications. First, several closely related works are not discussed [1,2,3,4]. As noted in [1,2], the nonparametric global solution of the reward-training loss does satisfy Pareto Optimality (PO). In practice, reward models are typically large language models (LLMs) implemented with transformer architectures and thus possess strong function-approximation capabilities. Consequently, practical reward models are much closer to expressive models than to the simplified linear or polynomial formulations considered in this paper. Hence, the extension from linear to polynomial reward functions has limited practical significance.

Second, Theorem 5.1 attempts to provide a positive result by showing that PO can be satisfied under certain conditions. However, the assumption of a uniform distribution is highly idealized and far from practical scenarios. Moreover, existing work do have some more practical positive results. They already indicate that when the reward model has sufficient approximation capacity, PO can in fact be satisfied.

Line 106: The statement “NLHF coincides with maximal lotteries” is missing an appropriate citation. The current reference (Fishburn, 1984) pertains to the definition of maximal lotteries, not to the connection between NLHF and maximal lotteries. Reference [3] establishes this equivalence and should be cited here. In addition, [4] applies these axioms to propose a new alignment approach.

Lastly, the English writing and representation should be improved. The current manuscript does not fully adhere to the ICLR style guidelines, and the use of fonts and formal structure is inconsistent with the official template.

[1] Ritesh Noothigattu, Dominik Peters, and Ariel D Procaccia. Axioms for learning from pairwise comparisons. Advances in Neural Information Processing Systems, 33:17745–17754, 2020.

[2] Jiancong Xiao, Zhekun Shi, Kaizhao Liu, Qi Long, Weijie J Su. Theoretical Tensions in RLHF: Reconciling Empirical Success with Inconsistencies in Social Choice Theory, 2025.

[3] Roberto-Rafael Maura-Rivero, Marc Lanctot, Francesco Visin, and Kate Larson. Jackpot! Alignment as a maximal lottery. arXiv preprint arXiv:2501.19266, 2025.

[4] K Kim, J Zhang, A Ozdaglar, PA Parrilo Population-Proportional Preference Learning from Human Feedback: An Axiomatic Approach, 2025.

**Questions:**

see weakness

---

> ### Author Response · Authors · 2025-11-24
>
> We thank the reviewer for their detailed review. Below we address the concerns mentioned by the reviewer.
>
> ## First concern ##
> > First, several closely related works are not discussed [1,2,3,4]. As noted in [1,2], the nonparametric global solution of the reward-training loss does satisfy Pareto Optimality (PO). In practice, reward models are typically large language models (LLMs) implemented with transformer architectures and thus possess strong function-approximation capabilities. Consequently, practical reward models are much closer to expressive models than to the simplified linear or polynomial formulations considered in this paper. Hence, the extension from linear to polynomial reward functions has limited practical significance.
> ### Regarding related work ###
>
> We thank the reviewer for pointing out related work. We will make sure to cite [1], which is a precursor of the paper by Ge et al. (which is the paper we build on). We agree that the non-parametric solution of the Bradley Terry loss satisfies PO. Indeed, this is a classic result and predates the work cited by the reviewer ([1] shows a weaker condition under which PO holds, and [2] appears to misstate PO) because Bradley–Terry (BT) scores induce the Borda ranking, see e.g. [5], and Borda (like essentially any reasonable rank-aggregation rule) satisfies PO [6].
>
> References [2] and [4] study alignment methods in finite, unstructured alternative spaces without parametrized reward models and thus are in a standard social choice setting where PO is generally easy to satisfy. Their work is thus only loosely related to our setting; their main connection to our work is via their discussion of Ge et al. and [1], and their mention of the axioms we use. We will now discuss [2] and [4] in our related work section in the main body of the paper.
>
> We also wish to point out that [2] and [4] appeared on arXiv less than four months before the ICLR submission deadline. According to the ICLR guidelines, authors are excused from not being aware of such contemporaneous work and from missing citations to unpublished work (such as [3]). We believe the absence of these citations in the original submission should not be counted against us.
>
> ### Regarding the significance of the polynomial impossibility result ###
> PO is a very minimal, simple axiom which is precisely why from our perspective its failure in the linear setting is so interesting. This is why our goal, from the outset, was to probe the boundary of Ge et al.'s impossibility result by isolating some of their assumptions that are less common in practice or that could be intervened on in practice to circumvent the impossibility. One such assumption was only fine-tuning a linear layer while keeping the backbone of the transformer frozen, as in practice models are more often fully fine-tuned. This motivated us to consider more expressive hypothesis classes for the reward model that are universal approximators. Here the reviewer’s grouping of linear and polynomial reward functions misses a crucial distinction: linear models are not universal approximators, whereas polynomial models are on compact domains, much like neural networks.
>
> Likewise, it is unclear to us what exactly the reviewer is referring to with “strong function-approximation capabilities of transformers.” If this is referring to universal-approximation theorems, those already apply to neural networks with suitable nonlinearities (even without transformer architecture), and as mentioned above also to polynomials.
>
> Crucially, the target reward function is unknown a priori, so the necessary width/depth (for neural networks) or degree (for polynomials) cannot be fixed in advance. Even if the hypothesis class is expressive enough in principle to fit any target reward function, in practice one must choose a finite size. Our theorem shows that, despite this nominal expressivity, there exists a simple dataset on which PO fails for a broad class of loss functions.
>
> Furthermore, we believe the study of polynomial reward functions is motivated by the following counter-factual: if it turned out that considering polynomial rewards is sufficient to guarantee PO, then this would be a significant insight that would have consequences for practice. Thus, we believe that mathematically ruling out this possibility is a worthwhile and a useful contribution. This point applies more generally to negative theoretical results.
>
> Finally, the new technical ideas we use to obtain our negative result for polynomial rewards will also prove useful for extending the result to more and more expressive classes of reward functions.

---

> ### Author Response · Authors · 2025-11-24
>
> That said, even the linear reward model assumption is not as unreasonable as the review suggests. “Linear probing” (LP), training a linear layer on top of frozen activations at some layer can be surprisingly useful in practice. For example, [11, 12] study using LP as a first stage before using it to fine-tune (“LP → FT”) and find settings where this outperforms full fine-tuning in out of distribution generalization. Moreover, recent LLM personalization work explicitly adopts simplified reward models, such as fine-tuning a single linear transformation while keeping the backbone frozen (e.g., [7]).
>
> ## Second concern ##
>
> > Second, Theorem 5.1 attempts to provide a positive result by showing that PO can be satisfied under certain conditions. However, the assumption of a uniform distribution is highly idealized and far from practical scenarios.
>
> We would like to clarify a potential misunderstanding. When we described our result as “idealized,” we were referring to the fact that we formulate it as an asymptotic / limit result (rather than, for example, stating an approximate uniform-coverage guarantee).
> Importantly, the unit length of the comparison directions can easily be enforced by a simple pre-processing step: just normalize the vectors from our sample to have length 1. Thus, the only real (data-set relevant) assumption in our positive theoretical result is that the samples uniformly cover all directions. For example, for pairs of principles any dataset that looks like a linear transformation of a cloud of points or rotationally symmetric points (Gaussian, disk,  or unions thereof etc) would suffice.
>
> > Moreover, existing work do have some more practical positive results. They already indicate that when the reward model has sufficient approximation capacity, PO can in fact be satisfied.
>
> As a general comment, many social choice for alignment papers (such as [2], [3], [4] mentioned by the reviewer) assume that every possible response is a “candidate” and that humans provide binary preferences over a finite, unstructured candidate set. This is an idealized assumption inherited from classical social choice, which typically works with finite candidate spaces. In the language/generative setting, the response space is enormous and this assumption does not map perfectly onto LLMs, which is why recent work such as [8] and Ge et al. has begun to move away from this assumption. Thus, unconstrained optimization/ the "non-parametric solution" in the reviewer’s sense, i.e., treating each possible text string as an independent free variable the model can tune, does not reflect how transformers actually operate. A transformer has a fixed number of shared parameters which is relatively small compared to the combinatorial number of possible outputs; these parameters define a single function that maps many different inputs into a shared representation space and then to output probabilities. Thus, transformers and neural networks more generally necessarily exploit structure and reuse features across texts (as studied, for example, in mechanistic interpretability), rather than assigning separate, unrelated parameters to each possible output. For this reason, we view the standard social choice assumption of a finite unstructured candidate set as a reasonable starting assumption, but, building on Ge et al., we move toward a setting that better reflects practical details such as text embeddings, model expressivity, and data filtering.
>
> ## Further comments ##
>
> > Line 106: The statement “NLHF coincides with maximal lotteries” is missing an appropriate citation. The current reference (Fishburn, 1984) pertains to the definition of maximal lotteries, not to the connection between NLHF and maximal lotteries. Reference [3] establishes this equivalence and should be cited here.
>
> Maximal Lotteries and NLHF are Nash equilibria of games and coincide because the corresponding payoff matrices are related by a positive affine transformation. We will clarify this in a footnote. Note that formally, NLHF is precisely the von Neumann winner defined in [14], defined before NLHF. The equivalence with maximal lotteries is explicitly pointed out by the authors in [15], and reference [3] from this year mentioned by the reviewer also points this out. We now cite all of these papers in the related work section, and have moved part of the related work to the appendix.
>
> > Lastly, the English writing and representation should be improved. The current manuscript does not fully adhere to the ICLR style guidelines, and the use of fonts and formal structure is inconsistent with the official template.
>
> We have made some improvements in our revision. We did not receive any concerns about presentation from other reviewers, so we would be very thankful for any concrete pointers on how to improve the presentation.

---

> ### Author Response · Authors · 2025-11-24
>
> ### References ###
> [1] Ritesh Noothigattu, Dominik Peters, and Ariel D Procaccia. Axioms for learning from pairwise comparisons. Advances in Neural Information Processing Systems, 33:17745–17754, 2020.
>
> [2] Jiancong Xiao, Zhekun Shi, Kaizhao Liu, Qi Long, Weijie J Su. Theoretical Tensions in RLHF: Reconciling Empirical Success with Inconsistencies in Social Choice Theory, 2025.
>
> [3] Roberto-Rafael Maura-Rivero, Marc Lanctot, Francesco Visin, and Kate Larson. Jackpot! Alignment as a maximal lottery. arXiv preprint arXiv:2501.19266, 2025.
>
> [4] K Kim, J Zhang, A Ozdaglar, PA Parrilo Population-Proportional Preference Learning from Human Feedback: An Axiomatic Approach, 2025.
>
> [5] Anderson, L. B., H. Dandurova, J. E. Falk, and L. Yeganova, “Relationships between Borda voting and Zermelo ranking,” Social Choice and Welfare, 32(3), 2009, pp. 355–365.
>
> [6] D. Farkas and S. Nitzan, “The Borda Rule and Pareto Stability: A Comment,” Econometrica, vol. 47, no. 5, 1979.
>
> [7] A. Bose, Z. Xiong, M. Fazel, Y. Chi, S. S. Du, and L. Xiao, “LoRe: Personalizing LLMs via Low-Rank Reward Modeling,” Conference on Language Modeling (COLM), 2025.
>
> [8] S. Fish, P. Gölz, D. C. Parkes, A. D. Procaccia, G. Rusak, I. Shapira, and M. Wüthrich, “Generative Social Choice,” in Proceedings of the 25th ACM Conference on Economics and Computation (EC ’24), 2024.
>
> [9] L. Bereska and E. Gavves, “Mechanistic Interpretability for AI Safety — A Review,” Transactions on Machine Learning Research (TMLR), 2024.
>
> [10] E. J. Hu, Y. Shen, P. Wallis, Z. Allen-Zhu, Y. Li, S. Wang, L. Wang, and W. Chen, “LoRA: Low-Rank Adaptation of Large Language Models,” in International Conference on Learning Representations (ICLR), 2022.
>
>
> [11] A. Tomihari and I. Sato, “Understanding Linear Probing then Fine-tuning Language Models from NTK Perspective,” in Advances in Neural Information Processing Systems (NeurIPS 2024), 2024.
>
> [12] A. Kumar, A. Raghunathan, R. Jones, T. Ma, and P. Liang, “Fine-tuning can distort pretrained features and underperform out-of-distribution,” in International Conference on Learning Representations (ICLR), 2022.
>
> [13] N. Rimsky, N. Gabrieli, J. Schulz, M. Tong, E. Hubinger, and A. M. Turner, “Steering Llama 2 via Contrastive Activation Addition,” in Proceedings of the 62nd Annual Meeting of the Association for Computational Linguistics (ACL 2024)
>
> [14] M. Dudík, K. Hofmann, R. E. Schapire, A. Slivkins, and M. Zoghi, “Contextual dueling bandits,” in Proceedings of the 28th Conference on Learning Theory (COLT), 2015
>
> [15] Y. Wang, Q. Liu, and C. Jin, “Is RLHF More Difficult than Standard RL? A Theoretical Perspective,” in Advances in Neural Information Processing Systems (NeurIPS), 2023.

---

### Comment · Area_Chair_hjXR · 2025-11-28
**Please Check the Authors' Responses**

Dear Reviewers,

The authors have posted their responses. Could you please take a moment to review their responses and check whether your concerns have been adequately addressed? If possible, kindly initiate the discussion at your earliest convenience.

Your timely assistance is essential for keeping the review process on track. Thank you very much for your support and contribution.

Best regards, Your AC

---

### Author Response · Authors · 2025-12-04

Dear Area Chair,

Given that the reviewers have not had the chance to update their reviews and scores this year, we provide a brief summary here of our rebuttal to help you reach a decision.

The second and fourth reviewer both recommend acceptance.

The third reviewer evaluates the paper as being marginally below acceptance. The four weaknesses (and corresponding four questions) mentioned in their review were already stated as avenues for future work in the conclusion of our paper (see Section 6). In our detailed response to this reviewer, we have given at least partial responses to all of these questions. In response to the reviewer's third question, we have added two new results about the Pairwise Majority Consistency (PMC) axiom (note that this also resolves the only weakness mentioned by Reviewer 4) in the revised version of our submission (in particular, see Theorem 5.2).

The first review is the most critical, with a reject recommendation. The reviewer's concerns are as follows:
- The reviewer mentions four related papers that should be discussed in our paper. Of those, two are very recent arXiv papers (so not citing them should not be counted against the submission according to ICLR guidelines) and one is not directly relevant. Moreover, we point out (with appropriate references) that two results mentioned by the reviewer are actually due to known results in prior work/folklore in social choice theory and the bandit literature and don't impact the significance of our results. In any case, we have updated our paper with all the references provided by the reviewer, explaining why some of them are not directly relevant.
- The reviewer claims that our result regarding polynomial reward functions (and also the result regarding linear reward functions by Ge et al.) has limited significance, because in practice transformers have "strong function-approximation capabilities". In our detailed response, we explain why we believe that our extension from linear rewards to polynomial rewards is significant in comparison to linear rewards not just technically but also with respect to practice (precisely because polynomials are universal approximators, similar to neural networks, and importantly unlike linear functions from prior work). We also explain why we don't believe that considering the non-parametric global solution as suggested by the reviewer (the solution of the unconstrained loss minimization with a free variable for each text/response in RLHF) is sufficient. Furthermore, we explain why it can even sometimes be reasonable to assume that rewards are linear in practice.
- The reviewer is concerned that the assumption for our positive result is too idealized and cannot be enforced in practice. In our detailed response, we argue that the first assumption (unit length comparison vectors) can easily be enforced by normalizing the vectors, while the second assumption (uniform coverage of directions with comparison vectors) holds e.g. if (a subset of) the data (that induces comparison vectors) comes from a Gaussian. Furthermore, the dataset can be designed so that the second assumption (approximately) holds by adding new datapoints in a careful way. See also our response to question 4 of the third reviewer for more on this last point.

Best regards,

The Authors

---

### Meta-Review · Area_Chair_o52D · 2026-01-13

**Summary:**

This paper studies when reward modeling for preference alignment satisfies fundamental social-choice axioms focusing on Pareto Optimality (PO). Built on the impossibility result from Ge et al. (2024) regarding PO of reward model training in linear social-choice model, the authors generalize it to impossibility for polynomial functions. The authors also provide a positive result showing that uniform directional coverage of preference data in the embedding space can restore PO guarantees for broad classes of loss-based rules.

The paper received mixed reviews, with two positive initial reviews (rating 8) and two negatives ones (4 and 2).  All reviewers find the theoretical results clear and technically solid. The paper extends Ge et al. (2024) to a practical setting, while can be used to guide future research on alignment  The main concerns include 1) lack of empirical validation (reviewer U5VA, ZJsy, KC58), 2) limited contribution compared to prior works (reviewer KC58), 3) idealized assumptions that may not hold in practice (reviewer KC58 and ZJsy), and 4) limited scope of axioms  which only focused on PO (reviewer ZJsy, 6CiF). The author rebuttal clarified some concerns. One issue to be noted is that reviewer KC58 pointed four papers for citation where two of them are recent arxiv papers. According to ICLR policy, authors are not required to compare to contemporaneous work or unpublished arxiv papers. So while it would be great to carefully compare with the suggested papers as they are indeed relevant, this issue is not a deciding factor in final recommendation. Overall this is a borderline paper. The recommedation is accept. The authors are suggested to take a substantial revision to compare with related works, and include empirical validation even in a small-scale if possible.

**Reviewer Concerns:**

Addressed concerns:

Idealized assumptions that may not hold in practice (reviewer KC58 and ZJsy): the authors explained the conditions under which the two assumptions may hold in practice.

Limited scope of axioms (reviewer ZJsy, 6CiF): The paper only focused on PO, and the authors addressed the concern by adding Theorem 5.2 on analysis for Pairwise Majority Consistency (PMC).

Llimited contribution compared to prior works (reviewer KC58): this is a valid concern and the authors clarified the relation to prior works and the significance of the result. As noted above, the authors are not required to compare to contemporaneous work or unpublished arxiv papers. It requires a careful revision to address in next version, but this issue is not a deciding factor in final recommendation and thus classified as addressed.

Outstanding concerns:

Lack of empirical validation (reviewer U5VA, ZJsy, KC58): the authors argued that empirical validation will be future research. It would be important to include even  a small-scale empirical study to validate the analysis. This requires a substantial revision to the paper and would make the claims stronger and more convincing to practitioners.

**Reviewer Scores:**

Reviewer U5VA (8), reviewer 6CiF (8): the reviewers will maintain positive ratings.

Reviewer ZJsy (4): The reviewers concern on limited scope of axioms is addressed with new analysis, while the concern of lack of empirical validation remains.

Reviewer KC58 (2): The reviewer likely will increase the score to 4 if not weigh in the concern on comparsion to the suggested arxiv papers.

---

### Decision · Program_Chairs · 2026-01-26

Accept (Poster)